# Acute Coronary Syndromes (ACS)—Unravelling Biology to Identify New Therapies—The Microcirculation as a Frontier for New Therapies in ACS

**DOI:** 10.3390/cells10092188

**Published:** 2021-08-25

**Authors:** Kaivan Vaidya, Bradley Tucker, Sanjay Patel, Martin K. C. Ng

**Affiliations:** 1Department of Cardiology, Royal Prince Alfred Hospital, Camperdown 2050, Australia; kaivan.vaidya@gmail.com (K.V.); sanjay.patel599@gmail.com (S.P.); 2Sydney Medical School, University of Sydney, Camperdown 2050, Australia; b.tucker@student.unsw.edu.au; 3Heart Research Institute, Newtown 2042, Australia; 4School of Medical Sciences, University of New South Wales, Kensington 2052, Australia

**Keywords:** microvascular obstruction, acute coronary syndrome, index of microcirculatory resistance, percutaneous coronary intervention, myocardial infarction

## Abstract

In acute coronary syndrome (ACS) patients, restoring epicardial culprit vessel patency and flow with percutaneous coronary intervention or coronary artery bypass grafting has been the mainstay of treatment for decades. However, there is an emerging understanding of the crucial role of coronary microcirculation in predicting infarct burden and subsequent left ventricular remodelling, and the prognostic significance of coronary microvascular obstruction (MVO) in mortality and morbidity. This review will elucidate the multifaceted and interconnected pathophysiological processes which underpin MVO in ACS, and the various diagnostic modalities as well as challenges, with a particular focus on the invasive but specific and reproducible index of microcirculatory resistance (IMR). Unfortunately, a multitude of purported therapeutic strategies to address this unmet need in cardiovascular care, outlined in this review, have so far been disappointing with conflicting results and a lack of hard clinical end-point benefit. There are however a number of exciting and novel future prospects in this field that will be evaluated over the coming years in large adequately powered clinical trials, and this review will briefly appraise these.

## 1. Introduction

In patients presenting with an acute coronary syndrome (ACS), protective mechanical and pharmacological interventions are aimed at infarct size reduction, reperfusion of viable myocardium, and attenuation of necrosis leading to heart failure [1]. Historically, the focus on ACS treatment has been on the epicardial coronary artery culprit lesion. In particular, primary percutaneous coronary intervention (PCI) is now the preferred reperfusion strategy for acute ST-elevation myocardial infarction (STEMI) and aims to restore patency of the epicardial infarct-related artery and achieve reperfusion to limit the extent of myocardial damage [2,3].

Advances in the treatment of STEMI patients has led to a progressive decline in mortality over several decades, with one-year cardiac mortality after primary PCI at 7–8% [4]. However, in approximately 50% of STEMI patients, effective reperfusion does not occur despite timely epicardial recanalisation due to the occurrence of coronary microvascular obstruction (MVO), which is a critical determinant of adverse cardiovascular outcomes including mortality and heart failure hospitalisation [5,6,7,8]. MVO is a key independent predictor of adverse left ventricular (LV) remodelling and major adverse cardiovascular events (MACE) irrespective of infarct size. Moreover, MVO may be a more potent predictor of MACE than infarct size [7,9] or patency of the infarct-related epicardial artery [5]. In a study of 249 STEMI patients over a median 6-year follow-up, those with CMR-demonstrated MVO had higher rates of MACE than those without (54% vs. 9%, *p* < 0.001) and MVO was the strongest predictor of MACE with incremental prognostic value over infarct size and LV ejection fraction [10].

The overwhelming evidence for the prognostic significance of MVO underpins the need for strategies to protect or restore coronary microcirculatory flow. To date, there has been a paucity of cardioprotective therapies targeting MVO. Moreover, the translation of potential strategies (such as reduction in thrombus burden, conditioning, and administration of anti-thrombotic or vasodilatory drugs) into improved clinical outcomes has been disappointing due to difficulties with clinical trial design and a lack of hard clinical end-point benefit [11]. In this review, we provide a detailed summary of the pathogenesis of MVO, diagnosis of MVO during evolving ACS, potential therapeutic strategies and their limitations, and an evaluation of future opportunities on the horizon to mitigate MVO in ACS patients.

## 2. Pathogenesis of MVO

MVO is defined as the inability to reperfuse the coronary microcirculation (microvessels < 200 µm diameter) in a previously ischaemic myocardial territory, despite restoration of epicardial vessel patency. There are several interdependent mechanisms underlying MVO and myocardial injury (Figure 1), which are delineated below [1,2,11].

### 2.1. Pre-Existing Coronary Microvascular Dysfunction

Coronary microvascular dysfunction (CMD) is a mismatch of myocardial blood supply and oxygen consumption due to coronary microvessel dysfunction and can occur with or without obstructive coronary artery disease (CAD) and/or myocardial diseases (Table 1). The key pathogenic mechanisms driving CMD include vascular remodelling and rarefaction, perivascular fibrosis, and functional alterations such as sympathetic activation, endothelial dysfunction or smooth muscle cell dysfunction [12,13].

Pre-existing CMD is often present in ACS patients before their index event, which can result in an increased risk of significant MVO and poor cardiovascular outcomes. Furthermore, pre-existent CMD likely represents an important pathological contributor of MVO, as studies have demonstrated that coronary flow is attenuated by up to 50% in non-culprit arteries during acute myocardial infarction (AMI), both before and after primary PCI, suggesting global rather than regional myocardial microcirculatory dysfunction [14]. Indeed, pre-existing CMD has been shown to be a strong independent risk factor for peri-PCI myocardial infarction in stable coronary heart disease patients, indicating that pre-existing CMD is a critical determinant of susceptibility for further microvascular injury [15]. In particular, Type 1 CMD is associated with a multitude of risk factors, many of which predispose to ACS itself, and which modulate functional and structural microvascular alterations. These risk factors include ageing, hypertension, diabetes mellitus, insulin resistance, dyslipidaemia, and chronic inflammation [1,2,12,13,16]. Chronic hyperglycaemia and insulin resistance in particular have been associated with significantly reduced coronary vasodilator function [17]. Moreover, acute hyperglycaemia also predisposes to MVO and reperfusion injury by causing microvascular leucocyte plugging, increasing platelet procoagulability, and elevating free fatty acid levels which reduce endothelial-dependent vasodilatation [2,18]. Similarly, hypercholesterolaemia impairs vascular endothelial function and reactivity by a reduction in nitric oxide bioavailability, elevated oxidative stress, and a proinflammatory milieu [19].

### 2.2. Individual Susceptibility

Various genetic factors also play a role in modulating adenosine-induced vasodilatation and subsequent microcirculatory function. The 1976T.C polymorphism of the adenosine 2A receptors gene is related to a higher prevalence of MVO, and genetic variations in regions of the VEGFA and CDKN2B-AS1 genes have been associated with CMD. Patients with MVO also have a more compact fibrin network, suggesting a genetically mediated resistance to clot permeability and lysis [1,2]. Finally, another key factor determining individual susceptibility to MVO is the presence of ischaemic pre-conditioning (IPC) and its clinical correlate pre-infarction angina, which is protective for both the myocardium and the coronary microcirculation [20].

### 2.3. Ischaemic Injury

Ischaemic injury is a well-described mechanism for cardiomyocyte cell death, especially if the period of myocardial ischaemia lasts over 3 h in duration. Ischaemia to the microcirculation results in capillary damage, endothelial protrusions/blebs which occlude the capillary lumen, decreased endothelial nitric oxide synthase (eNOS) vasodilator release, and increased endothelin-1 vasoconstrictor release. Furthermore, ischaemia induces disruption of the endothelial glycocalyx barrier and destabilization of interendothelial cell junctions resulting in extravascular erythrocytes, adhesion and influx of inflammatory cells, and interstitial myocardial oedema which compresses capillaries and small arterioles, further reducing flow through these dysfunctional vessels in a vicious cycle [1,2,3,16]. If there is no reperfusion of the infarct-related artery, the eventual outcome is irreversible necrosis and complete MVO of the affected myocardial territory, although collateral circulation can be somewhat protective in this process.

### 2.4. Ischaemia-Reperfusion Injury

After a prolonged period of ischaemia, reperfusion of the affected region potentiates further injury to the microvasculature. Reperfusion stimulates reactive oxygen species (ROS) production by mitochondria, which subsequently results in increased membrane permeability, calcium overload, endothelial swelling, and cell disruption. An influx of leucocytes, platelets, and neutrophil-platelet aggregates after reperfusion exacerbates vessel lumen obliteration, coronary microvascular constriction, oxidative stress injury, and release of proteolytic enzymes and pro-inflammatory mediators which drive further MVO [1,2,3,16]. Pericytes that cover the coronary capillaries also contract during post-ischaemia reperfusion, contributing to the reduction in microvascular blood flow. Myocardial interstitial oedema is also enhanced in early reperfusion by the washout of osmotically active mediators from the intravascular space during reactive hyperaemia. The most severe manifestation of coronary microcirculatory injury due to reperfusion is intramyocardial haemorrhage (IMH)—after substantial endothelial disruption, erythrocyte extravasation into the perivascular space, and inflammatory and coagulation cascade activation resulting in thrombosis and coagulation factor consumption. IMH-induced interstitial iron deposition is visible on cardiac magnetic resonance imaging (CMR) T2-weighted and T2*-weighted sequences in 40%–80% of patients with MVO [5,21], and can induce an inflammatory response to further intensify MVO. Importantly, patients with IMH and MVO have a worse outcome than those with MVO without IMH [5].

### 2.5. Distal Microembolisation

Distal coronary microembolisation is the final key mechanism underpinning myocardial injury and MVO. The spontaneous or iatrogenic (PCI-induced) rupture of atherosclerotic epicardial plaque releases particulate debris which, together with superimposed microthrombi, embolises distally into the coronary microcirculation. This microvascular plugging by inflamed plaque fragments contributes to MVO (represented as the coronary no-reflow phenomenon), and subsequent peri-procedural MI. In STEMI, distal embolization of thrombotic microemboli and atherothrombotic debris has been increasingly recognised as an important mechanism contributing to coronary microcirculatory impairment. In an extensive microscopic examination of 25 cases of sudden death due to acute coronary thrombosis, Falk provided conclusive support for spontaneous, mainly thrombotic, distal embolization leading to occlusion of small intramyocardial arteries and microinfarctions [22]. Furthermore, microvascular fibrin and platelet aggregates are found in patients dying of MI [23].

Coronary no-reflow describes the inability to reperfuse regions of previously ischaemic myocardium after the restoration of flow in the occluded coronary artery, and angiographically can be visualised as impaired penetration of contrast distally into the myocardium [11,16]. It is important to note however that distal embolization, whether by iatrogenic rupture of an atherosclerotic plaque via angioplasty or spontaneous plaque erosion during the ACS, releases both particulate debris as well as soluble factors which impair microvascular perfusion but cannot be captured with filter devices or meshes [3,24]. This process is believed to be mediated in part by activated neutrophils in vulnerable coronary plaque, and enhanced release of neutrophil-derived mediators such as neutrophil extracellular traps (NETs) and microparticles which promote pro-thrombotic and pro-inflammatory cascades via neutrophil-platelet aggregate formation, complement cascade activation, inflammatory cytokine release from activated endothelial cells, ROS generation causing oxidative damage, and local tissue degradation [25,26,27,28]. This cascade is further augmented by angioplasty-induced plaque rupture and exposure of tissue factors which promotes thrombosis and subsequent MVO [29]. In STEMI patients post-primary PCI, coronary NETs burden correlates positively with infarct size and negatively with ST-elevation resolution [28].

### 2.6. Inflammation

Inflammatory activation is triggered by myocardial ischaemia and necrosis; mediating tissue healing, scar formation and ventricular remodelling, and serving as a potent contributor to cardiac MVO [30,31]. After the acute phase of a myocardial infarction, a robust systemic inflammatory response is driven primarily by the infiltration and activation of neutrophils, followed by monocyte/macrophages and lymphocytes. This response, aiming to remove cellular debris and repair injured tissue, is mediated by the release of various cytokines and acute-phase proteins [31,32]. In particular, cellular debris activates the NLRP3 inflammasome—a macromolecular protein complex that regulates caspase 1 activation and subsequent production of potent pro-inflammatory cytokines IL-1ß and IL-18 [31,33]. The inflammasome potentiates the inflammatory response in ACS by amplifying tissue injury and leading to interstitial myocardial oedema, which compresses capillaries and arterioles in the coronary microcirculation, causes arteriolar spasm and other progressive endothelial abnormalities, and results in erythrocyte stasis and micothrombosis within the microvasculature, all of which contribute to MVO [34,35]. Produced downstream of the inflammasome, IL-6 induces C-reactive protein (CRP) synthesis and acts as a secondary mediator of the inflammatory cascade [33]. There is growing evidence to suggest that CRP is not only a marker of atheroinflammation and cardiovascular risk, but is also a mechanistic proinflammatory mediator of myocardial damage and MVO. Experimental studies [32,34,35] have demonstrated that CRP promotes myocyte apoptosis, increases infarct size, and impairs infarct healing in a rat model; and overexpression exacerbates ventricular remodelling after myocardial infarction. In a multitude of human clinical studies [32,35,36,37,38,39,40], elevated CRP levels post-myocardial infarction have been shown to be associated with larger infarct size, development of heart failure, adverse LV remodelling, and MVO as evaluated by CMR. CRP induces myocardial and endothelial injury via direct and indirect stimulation of the coagulation and complement cascade, particularly by altering the fibrinolytic balance of endothelial cells and promoting intravascular fibrin formation, and it is postulated that this process contributes to MVO in ACS patients [34,40].

## 3. Diagnosis of MVO during Evolving ACS

The diagnosis of MVO is made using both non-invasive and invasive tools, with incidence rates ranging from 10% angiographically to 60% using CMR [1,2]. The discrepancy in MVO diagnosis between various modalities can be attributed to the fundamental differences in assessing functional alterations in coronary blood flow and structural changes at a tissue level. Furthermore, MVO is a dynamic process with the potential for reversible and irreversible damage (approximately 50% in each cohort) [41]. Therefore, the timing of evaluation (early after reperfusion vs. late via CMR) is a key factor in determining microcirculatory dysfunction [11] in an evolving ACS setting. Importantly, most studies of MVO have been limited to STEMI patients with the few studies in non-ST elevation myocardial infarction (NSTEMI) patients limited by small patient sizes. A larger study of 190 NSTEMI patients [42] reported an MVO incidence rate of 13.8% using CMR and established culprit lesion and infarct size as independent predictors of MVO.

### 3.1. Invasive Indices

aAngiography

Thrombolysis in myocardial infarction (TIMI) flow is a widely recognised angiographic visual scoring system from 0 to 3 grading epicardial coronary flow, ranging from no flow to complete antegrade flow to the distal vessel which is associated with less MVO, smaller infarct size, fewer MACE, and better survival [1,16]. Although fast and easy to use, over half of all patients with TIMI 3 flow demonstrate ongoing CMR-determined MVO, thereby limiting its sensitivity and predictive power [43]. A variation on this approach is the corrected TIMI frame count (cTFC) which provides a quantitative index to evaluate coronary flow by counting the number of frames required for contrast medium to reach a standardised landmark, although its reproducibility outside of core labs and prognostic value is not clear [1,16]. Myocardial blush grade (MBG) and TIMI MBG are two other methods developed to evaluate the kinetics of dye penetration into the myocardium, shifting attention away from epicardial coronary flow to the microcirculation. MBG is a densitometric method evaluating myocardial contrast blush (and hence perfusion) after injection, ranging again from 0 to 3 [3,16]. It provides additional diagnostic value to TIMI flow as in patients with TIMI 3 flow, two-thirds demonstrated MBG grade 0 to 1; and it has important prognostic value independent of TIMI flow [44]. However, once again, preserved MBG does not necessarily confirm sufficient microvascular perfusion, and a large subset of patients with MBG grade 2–3 show CMR-determined MVO [43,45].

bCoronary physiology

Invasive assessments of coronary epicardial and microvascular function can be performed with an intracoronary sensor-tipped guidewire [46]. Coronary physiologic indices proposed for assessment of the coronary microcirculatory function include coronary flow reserve (CFR), index of microcirculatory resistance (IMR), hyperaemic microvascular resistance (HMV), and resistive reserve ratio (RRR) [16]. CFR interrogates the epicardial and microvascular coronary circulation and can be derived using Doppler or thermodilution techniques, as either the ratio of hyperaemic blood flow divided by resting flow or mean transit time (Tmn) at rest divided by Tmn during hyperaemia. A CFR value < 2.0 has a sensitivity of 79% and specificity of 34% for CMR-determined MVO, and CFR of the infarct-related artery portends prognostic value such as LV function post-MI and long term mortality [16,47]. However, CFR is limited by poor reproducibility and variations in measurement depending on systemic haemodynamic conditions such as heart rate, blood pressure, and left ventricular contractility [48]. IMR, on the other hand, is a more specific and reproducible measure of the microcirculation independent of haemodynamic parameters or the severity of epicardial stenoses [49,50]. IMR, measured using a pressure-temperature sensor guidewire under peak hyperaemia (induced via adenosine infusion), after three intracoronary injections of room temperature saline, is equal to the hyperaemia distal arterial pressure (P_d_) multiplied by the hyperaemic Tmn (IMR = P_d_ × Tmn). IMR is an appealing diagnostic technique as it can be measured immediately during primary PCI and thereby identify high-risk patients requiring more intensive follow-up or early intervention with novel therapies targeting MVO and aimed at microvascular recovery. A meta-analysis found that IMR is significantly higher in patients with CMR-determined MVO [51]. In STEMI patients, IMR was an independent predictor of myocardial salvage (13% decrease in salvage for every 20% rise in IMR) and LV ejection fraction (*p* < 0.01). IMR has also been shown to predict infarct size, 3-month echocardiographic wall motion score, and peak creatinine kinase and troponin levels better than TIMI grade or CFR [50,52,53]. In a landmark study of 253 STEMI patients, IMR > 40 (which occurred in 32% of patients) was a powerful independent predictor of the primary endpoint (death and heart failure hospitalisation) at one year (17.1% vs. 6.6%, *p* = 0.027) and at three years (20% vs. 11%, *p* = 0.04) compared to those with IMR ≤ 40 [8]. HMV is calculated as distal pressure divided by the mean Doppler flow velocity at peak hyperaemia using a coronary guidewire with a pressure sensor and Doppler transducer, with a value > 2.5 mmHg cm^−1^ s indicative of CMR-determined MVO (sensitivity 71%, specificity 63%) [16]. Although there is some data associating higher HMV with poor ventricular recovery and higher MACE (death and heart failure rehospitalization) [54,55], its role in evolving ACS has been studied less than IMR and a major limitation is the difficulty in acquiring high-quality measurements using the Doppler wire. Finally, the RRR measures the ability to achieve maximal coronary hyperaemia and quantifies the vasodilator response of the coronary microcirculation as the ratio of basal resistance (without adenosine) to IMR at peak hyperaemia, with lower values seen in STEMI patients [16].

cIntracoronary ECG

Intracoronary ECG has also been proposed as an early, simple and objective method of diagnosing MVO and predicting infarct zone recovery “real-time” during primary PCI, thereby identifying patients suitable for potential therapeutic interventions or careful follow-up [56]. In a study of 64 STEMI patients undergoing primary PCI [57], intracoronary ST-segment resolution (≥1 mm improvement compared to baseline) correlated with MVO (*p* = 0.005) on contrast-enhanced CMR assessed day 4 post-STEMI; as well as infarct size, non-viable mass, peak creatinine-kinase and left ventricular remodelling at 3 months post-STEMI.

### 3.2. Non-Invasive Indices

After primary PCI, incomplete ST-elevation resolution has been related to MVO and portends a worse clinical outcome [58], although this modality lacks specificity or quantitative ability, and is quite dynamic during an evolving ACS. Myocardial contrast echocardiography utilizes ultrasound to visualise contrast microbubbles with a rheology akin to red blood cells, and lack of intra-myocardial contrast opacification due to MVO is predictive of left ventricular remodelling post-STEMI [1]. However, again, this modality is limited by spatial resolution, operator dependence, semi-quantitative assessment, and incomplete LV coverage with suboptimal visualisation.

Myocardial perfusion positron emission tomography (PET), based on radiotracers labelled with isotope-emitting positrons, represents a well-validated imaging modality for the identification of CMD with high diagnostic accuracy, temporal resolution, and sensitivity. Rest and stress PET allows for quantification of indices of microvascular dysfunction such as absolute myocardial blood flow (MBF), myocardial perfusion reserve (MBF at maximal stress), and coronary flow reserve (ratio of MBF during maximal coronary vasodilatation to resting MBF) [59,60]. Hybrid positron emission tomography-computed tomography (PET-CT) has also been utilised for MVO evaluation, by measuring regional 18F-deoxyglucose (FDG) uptake, with promising results as it allows for quantification of myocardial tissue inflammation after reperfusion [61].

Finally, CMR is the gold standard non-invasive diagnostic modality for MVO, allowing multislice imaging with high tissue contrast and spatial resolution, enabling quantification and localization of MVO and infarct size, as well as regional contractile function. On contrast-enhanced CMR, MVO is identified as a dark hypointense core (lack of gadolinium enhancement) during first pass (<2 min) or after conventional late (10–15 min) gadolinium enhancement of the affected myocardium. First pass (early) MVO is more sensitive than late MVO to subtle microvascular injury, as the latter often underestimates severity. However, the prognostic utility of early MVO for the prediction of postinfarction MACE is lower, as late MVO represents severely disturbed microcirculation and hence correlates more strongly with worse outcomes [1,2,43,62,63]. Newer techniques in CMR further facilitate infarct characterization, identification of IMH, and quantification of oedema and myocyte loss [16]. A key point of variability between CMR evaluation of MVO is the timing of image acquisition due to the dynamic course of microvascular injury, and this has led to standardised protocols for postinfarction CMR imaging to ensure the validity of the acquired data [2].

## 4. Therapeutic Approaches in MVO

Over the course of several years, many therapies have been trailed to prevent as well as treat MVO, and this represents an ongoing unmet need in interventional cardiology. However, unfortunately, there still exists no treatment which has convincingly demonstrated benefit in an adequately powered randomised control trial with clinical endpoints. Regardless, we will summarise various treatments which have been investigated as well as possible future directions in the field.

### 4.1. Beta Blockers

Pre-clinical data has shown that third-generation beta-blockers (carvedilol, nebivolol) may reduce infarct size and protect the coronary microcirculation [2]. Intravenous administration of metoprolol before reperfusion in an animal model has been shown to reduce infarct size and MVO, by modulating the inflammatory response inhibiting neutrophil-platelet aggregates [64]. In a study (METOCARD-CNIC) of 270 patients with anterior STEMI, ambulance intravenous metoprolol (3 × 5 mg) administered pre-PCI reduced infarct size, prevent LV adverse remodelling, preserved LV systolic function, and reduced future heart failure readmissions [65]. A subanalysis of this trial [66] demonstrated that in control patients, there was a positive correlation between neutrophil count and extent of MVO, whereas in the treatment arm this was not the case, suggesting that administration of metoprolol during ongoing ACS does not affect circulating neutrophil levels but in fact modulates the effect of neutrophils on MVO. However, another study [67] (EARLY BAMI) of 683 patients failed to demonstrate any reduction in infarct size (CMR-determined) in STEMI patients given intravenous metoprolol (2 × 5 mg) just before primary PCI.

### 4.2. Statins

Several studies have proposed that acute statin treatment may mitigate MVO via pleiotropic effects such as improvement in endothelial function, coronary microvascular dilatation, and anti-inflammatory and anti-thrombotic mechanisms. However, the evidence regarding this remains conflicting. Chronic ongoing statin therapy at the time of STEMI is associated with a lower rate of coronary no-reflow, and more favourable LV ejection fraction and ventricular remodelling [68]. In 171 STEMI patients (STATIN STEMI) [69], high-dose atorvastatin pre-treatment did not reduce MACE but did improve coronary flow after primary PCI, as assessed angiographically by cTFC and MBG. In the SECURE-PCI trial, 4191 patients with ACS were randomised to receive two loading doses of 80 mg atorvastatin or placebo before and 24 h after planned PCI, followed by 40 mg daily thereafter. At 30 days, however, rates of MACE were not lower in the treatment arm, although in the subgroup of 1012 STEMI patients the difference was statistically significant in favour of the treatment cohort (8.5% vs. 12.6%, hazard ratio [HR] 0.66). Finally, a smaller study [70] of STEMI patients given 80 mg atorvastatin pre-PCI and for 5 thereafter versus controls (10 mg daily post-PCI) did not improve infarct size measured by technetium Tc99m single-photon emission computed tomography, MBG, or ST-segment elevation resolution.

### 4.3. Adenosine

Adenosine is a potent direct vasodilator of the coronary microcirculation and also has a range of purported pleiotropic effects including anti-inflammatory properties against neutrophils, inhibition of platelet aggregation, antiapoptotic effects, and stimulation of angiogenesis [71]. However, clinical studies have displayed mixed results in attenuation of MVO and infarct size. In the AMISTAD-II study [72,73] of over 2000 anterior STEMI patients receiving thrombolysis or primary PCI, patients were randomised to a 3-h adenosine infusion or placebo. Here, clinical outcomes were not improved overall in the adenosine arm but infarct size as assessed by technetium-99m sestamibi tomography was reduced in patients who received 70 mcg/kg/min adenosine dosing. Furthermore, in a post hoc analysis, adenosine administered within the first 3.2 h of evolving anterior STEMI (from chest pain onset) enhanced 1-month and 6-month survival compared to placebo, and the composite endpoint of death or heart failure at 6 months. In the REOPEN-AMI trial [74,75] of 240 STEMI patients treated with primary PCI and thrombus aspiration, the additional intracoronary administration of adenosine (but not nitroprusside) resulted in a significant improvement in MVO as assessed by ST-elevation resolution (51% vs. 71%, *p* = 0.009) but not TIMI flow grade and MBG (18% vs. 30%, *p* = 0.06) or MACE at 30 days (10% vs. 20%, *p* = 0.08) compared to placebo. At one-year follow-up, however, the adenosine cohort had reduced MACE and better LV remodelling compared to placebo or the nitroprusside arm. In contrast, however, the REFLO-STEMI trial [76] of 247 STEMI patients failed to confirm these findings, and in fact, demonstrated a similar extent of MVO but larger infarct size, lower LV ejection fraction, and higher rates of MACE at 30 days and 6 months in the high dose intracoronary adenosine cohort compared to placebo.

### 4.4. Atrial Natriuretic Peptide

Atrial natriuretic peptide (ANP) suppresses endothelin-1 production and acts on the reperfusion injury salvage kinase (RISK) cardioprotective pathway which is involved in ischaemic preconditioning, thereby exerting possible favourable effects on MVO [77]. In the J-WIND trial, carperitide (ANP agonist) treatment resulted in a reduction in enzymatic infarct size and improved LV ejection fraction in STEMI patients treated pre-PCI compared to placebo. However, there remains a paucity of clinical study data evaluating the effect of ANP on MVO.

### 4.5. Exenatide

Exenatide, a glucagon-like peptide-1 agonist, has been proposed to reduce infarct size and preserve cardiac function in pre-clinical models, by reducing myocardial apoptosis and oxidative stress. Unfortunately, human trials have been limited by conflicting results. In an initial study of 172 STEMI patients [78], the exenatide-treated patients demonstrated larger myocardial salvage index and smaller infarct size (*p* = 0.003) on CMR compared to controls, and this was further reinforced in a smaller study of 58 patients where the absolute mass of delayed hyperenhancement (suggestive of MVO on CMR) in exenatide-treated patients was significantly reduced compared to controls (*p* < 0.01) [79]. However, a subsequent larger study of 191 patients showed no difference between the two groups (exenatide infusion vs. placebo) in terms of infarct size on CMR corrected for the area at risk (*p* = 0.662) [80].

### 4.6. Antiplatelet Therapy

Current practise guidelines [81] recommend (Class IIA, evidence level C) GP IIb/IIIa inhibitor therapy (abciximab, tirofiban, or eptifibatide) in patients with slow or no-reflow, given either intravenously or intracoronary initially as a bolus followed by a maintenance infusion, although the evidence remains conflicting. In the FINESSE trial [82] of 2452 STEMI patients, upstream abciximab administration with half-dose reteplase significantly reduced infarct size but not MACE or mortality at 90 days or one year. In 110 patients with acute myocardial infarction, however, early abciximab administration did not lead to smaller CMR-determined infarct size at 6 months overall except in patients with a longer transportation time [83]. In the RELAX-AMI trial [84] of 210 STEMI patients treated with primary PCI, early abciximab therapy improved pre-PCI angiographic findings (TIMI flow grade, cTFC, MBG), post-PCI tissue perfusion (ST-elevation resolution and MBG), and one-month LV function recovery (*p* = 0.03) compared to later therapy in the catheterisation lab. The ON-TIME 2 trial [85] of 984 STEMI patients demonstrated that routine pre-hospital initiation of high-dose tirofiban improved ST-elevation resolution and clinical outcomes post-PCI. In the INFUSE-AMI study [86] of 452 anterior STEMI patients, infarct size and mass (*p* = 0.03) measured with CMR at 30 days was reduced by bolus intracoronary abciximab delivered to the infarct lesion site compared to placebo. Finally, in AIDA STEMI [87], the only trial powered for clinical outcomes with 2065 STEMI patients, there was no difference between intracoronary and intravenous administration of abciximab in terms of MACE and MVO rate (CMR-assessed) at follow-up.

Experimental data has also examined the role of P2Y12 inhibitors in reducing MVO and infarct size when administered at the onset of reperfusion or earlier. In a small study of 84 STEMI patients who received upstream clopidogrel treatment prior to PCI, there was a significantly lower rate of MVO on CMR (odds ratio [OR] 0.39, *p* = 0.002) [88]. However, a subanalysis of the PLATO trial comparing ticagrelor and clopidogrel did not find any difference in myocardial perfusion or coronary flow as assessed by TIMI flow grade or MBG [89]. In the ATLANTIC study of 1862 STEMI patients [90], pre-hospital administration of ticagrelor did not improve pre-PCI coronary reperfusion. The REDUCE-MVI trial [91] of 110 STEMI patients found no difference between ticagrelor and prasugrel in reducing MVO or infarct size, as assessed by invasive IMR evaluation and CMR at one month. The ongoing PITRI trial will aim to determine whether intravenous cangrelor administered prior to reperfusion reduces infarct size or MVO as assessed by CMR [92].

### 4.7. Ischaemic Conditioning

Ischaemic conditioning, defined by either repeated brief episodes of mechanical ischaemia/reperfusion before PCI or at the onset of reperfusion, has been proposed as a cardioprotective therapy in several clinical studies, by increasing myocyte resistance to ischaemic injury. Post-conditioning refers to interrupted reperfusion with alternating brief coronary artery re-occlusion-reperfusion cycles. In a study of 118 STEMI patients, patients in the treatment arm demonstrated better ST-elevation resolution and smaller CMR-determined infarct size, but no difference in clinical outcomes at 15 months [93]. In a trial of 122 STEMI patients [94], post-conditioning did not improve infarct size, myocardial salvage, or LV ejection fraction two days post-PCI, but at 12 months, CMR demonstrated more favourable remodelling (*p* < 0.03) and less MVO (*p* = 0.05) in the therapy cohort. In the much larger POST trial of 700 STEMI patients, however, 4 cycles of 60-s angioplasty balloon inflation-deflation did not improve myocardial reperfusion (MBG), ST-elevation resolution, or MACE at 30 days [95]. Finally, the large multicentre DANAMI-3-iPOST trial of 1234 STEMI patients [96], involving four repeated 30-s cycles before stent implantation, found no difference in MACE (HR 0.93, *p* = 0.66), or CMR-evaluated infarct size, myocardial salvage index, the extent of MVO, or LV ejection fraction at 3 months.

On the other hand, remote ischaemic post-conditioning (RIPC), using several cycles of brief limb ischaemia and reperfusion, has shown some promise in smaller studies but not larger trials powered for hard outcomes. In the CONDI trial of 333 STEMI patients, four cycles of 5-min arm (blood pressure cuff) ischaemia alternating with reperfusion during ambulance transport pre-PCI improved myocardial salvage (*p* = 0.03) and hence infarct size as assessed by myocardial perfusion imaging [97]. This was further reinforced in a smaller study of 100 anterior STEMI patients, where 3-cycle lower limb RIPC at the time of primary PCI reduced enzymatic infarct size, tissue oedema volume (on CMR), and improved ST-elevation resolution [98]. Similarly, a study of 197 STEMI patients receiving 4-cycle upper arm RIPC prior to primary PCI demonstrated reduced CMR-assessed infarct size, enhanced myocardial salvage, and less tissue oedema in the treatment cohort [99]. In the LIPSIA CONDITIONING trial of 696 STEMI patients, post-conditioning alone with four 30-s re-occlusion/reperfusion cycles did not improve myocardial salvage or MVO as assessed by CMR, but when combined with RIPC (three 5-min upper arm cycles) it improved myocardial salvage [100], translating into reduced MACE (OR 0.56, *p* = 0.04) and heart failure (OR 0.32, *p* = 0.02) at over three years follow-up [101]. Finally, however, the very large CONDI-2/ERIC-PPCI trial of 5401 STEMI patients [102], which compared standard care including sham simulated RIPC to four 5-min upper arm RIPC cycles before PCI, found no difference in MACE (HR 1.1, *p* = 0.32), i.e., cardiac death or heart failure hospitalisation, at 12 months.

### 4.8. Interventional Procedures

aAspiration thrombectomy

Initial smaller studies evaluating manual thrombus aspiration, purported to improve myocardial perfusion and attenuate MVO by reducing distal embolisation of atherothrombotic debris, showed promising results but this did not translate into a benefit in larger trials. In the small REMEDIA trial of 100 STEMI patients, manual aspiration improved MBG (OR 2.6, *p* = 0.02) and ST-elevation resolution (OR 2.4, *p* = 0.034) [103]. Similarly, the TAPAS study of 1071 patients demonstrated better MBG (relative risk [RR] 0.65, *p* < 0.001) and ST-elevation resolution (RR 1.28, *p* < 0.001) [104] in the treatment cohort. At one year follow-up, this translated into less cardiac mortality and non-fatal reinfarction (HR 1.81, *p* = 0.009) for patients receiving aspiration thrombectomy [105]. In a meta-analysis of nine trials with 2417 patients, adjunctive aspiration thrombectomy was associated with better epicardial TIMI flow grade perfusion (*p* < 0.0001), post-procedural MBG (*p* < 0.0001), less distal embolisation (*p* < 0.0001), and significant 30-day mortality benefit (1.7% vs. 3.4%, *p* = 0.04) [106].

However, in the much larger TASTE trial of 7244 STEMI patients, there was no difference at 30 days between the aspiration thrombectomy arm and controls in all-cause mortality (HR 0.94, *p* = 0.63), hospitalisation for recurrent myocardial infarction (HR 0.61, *p* = 0.09), or stent thrombosis (HR 0.47, *p* = 0.06) [107]. These findings were reaffirmed at one year follow-up as well for all-cause mortality (HR 0.94, *p* = 0.57), recurrent myocardial infarction hospitalisation (HR 0.97, *p* = 0.81), and stent thrombosis (HR 0.84, *p* = 0.51) [108]. Similarly, in the TOTAL trial of 10732 STEMI patients, there was no difference in the primary outcome (cardiovascular death, recurrent myocardial infarction, cardiogenic shock, or New York Heart Association [NYHA] Class IV heart failure; HR 0.99, *p* = 0.86) but stroke within 30 days was higher in the thrombectomy arm compared to controls (HR 2.06, *p* = 0.02) [109]. On the other hand, in patients with angiographic evidence of a large thrombus burden, the use of the Angiojet mechanical thrombectomy device in the JETSTENT study of 501 STEMI patients was associated with improved ST-elevation resolution (*p* = 0.043) and reduced MACE (*p* = 0.011) at 6 months [110]. In the MASTER trial of 433 STEMI patients, patients were randomised to either conventional stent implantation or a novel polyethylene terephthalate micronet mesh-covered stent, designed to trap and exclude thrombus and friable atheromatous debris, to prevent distal embolisation [111]. Here, the MGuard stent improved ST-elevation resolution (*p* = 0.008) and TIMI flow grade (*p* = 0.006) but not MBG (*p* = 0.81), mortality (*p* = 0.06) or MACE (*p* = 0.75) at 30 days.

bDeferred coronary stenting in STEMI

Importantly, there has also been an examination of whether deferred stenting (after successful reperfusion with initial thrombectomy and/or balloon angioplasty) might reduce no-reflow and salvage myocardium in high-risk STEMI patients, particularly in the DEFER-STEMI pilot trial [112]. Comparing immediate stenting to deferred (intention-to-treat) stenting 4–16 h later in selected patients at risk of no-reflow, fewer patients in the deferred cohort had no-/slow-reflow based on TIMI flow grade (OR 0.16, *p* = 0.006) and intraprocedural thrombotic events (*p* = 0.01). Of the 52 patients randomised to deferred stenting, recurrent STEMI occurred in 2 patients before the planned second procedure. Myocardial salvage index at 6 months was also lower in the deferred cohort compared to controls (*p* = 0.031), but there was no difference in CMR-assessed MVO at two days (*p* = 0.155).

cIntermittent Coronary Sinus Occlusion

Pressure-controlled intermittent coronary sinus occlusion (PICSO) has been proposed as a means of improving microvascular perfusion and reducing MVO, by redistributing venous blood to the border zone of ischaemic myocardium, thereby enhancing washout of injurious agents in the microcirculation and inducing the release of vascular endothelial growth factors [2]. In the OxAMI-PICSO observational study of 105 STEMI patients, selected patients treated with PICSO (if pre-PCI IMR > 40) had a lower IMR at 24–48 h (24.8 vs. 45, *p* < 0.001) and lower infarct size on CMR at 6 months (26% vs. 33%, *p* = 0.006) [113]. There is however no randomised data supporting its routine use.

### 4.9. Intracoronary Thrombolysis

There is a growing body of evidence supporting the hypothesis that intracoronary thrombolysis improves microvascular perfusion in STEMI patients, by induction of fibrinolysis and inhibition of red cell and platelet aggregation contributing to MVO. In a landmark study, 41 STEMI patients were randomized to low-dose intracoronary streptokinase (STK) (250 kU) immediately post-PCI or no additional therapy [114]. Intracoronary STK was associated with improved measures of microvascular perfusion, including IMR (11.7 in STK arm vs. 29.1 in controls, *p* < 0.001), CFR (2.3 vs. 1.7, *p* = 0.002) and TIMI frame count (19.1 vs. 27.5, *p* = 0.001). In a subsequent larger study of 95 patients by the same authors [115], patients receiving low-dose intracoronary STK had better two-day invasive microvascular function indices including CFR (2.5 vs. 1.7, *p* < 0.001) and IMR (20.2 vs. 34.2, *p* < 0.001) that correlated with reduction in infarct size (22.7% vs. 32.9%, *p* = 0.003) and improvement in left ventricular ejection fraction (57.2% vs. 51.8%, *p* = 0.018) at 6 months. However, another recent study of 76 anterior STEMI patients compared the effects of intracoronary tenecteplase to abciximab during primary PCI, and found better angiographic parameters including cTFC (14.1 vs. 18.2, *p* = 0.02) and TIMI flow grade (*p* = 0.03) in the abciximab cohort two days later, as well as no difference in infarct size on CMR at four months follow-up (*p* = 0.33) [116,117]. Furthermore, the T-TIME study of 440 randomised patients, evaluating low-dose intracoronary alteplase (10 or 20 mg) infused early after coronary reperfusion and before stenting to attenuate MVO in the STEMI population [118], was stopped early due to futility as there was no difference in CMR-assessed MVO (%left ventricular mass) between alteplase 20-mg and placebo (3.5% vs. 2.3%, *p* = 0.32) or 10-mg and placebo (2.6% vs. 2.3%, *p* = 0.74). Interestingly, higher prothrombin concentrations in the alteplase group suggested that the undesired procoagulant effect of fibrinolytic therapy via thrombin activation may have contributed to microvascular thrombosis, especially in patients with inadequate therapeutic anticoagulation with unfractionated heparin.

Two ongoing trials will stratify enrolment and patient selection based on IMR after primary PCI, hoping to identify the specific STEMI patient population most at risk of clinically significant MVO and therefore most likely to derive benefit from putative therapies. These two trials are the OPTIMAL study (www.clinicaltrials.gov; Unique Identifier NCT02894138) of 80 patients (IMR > 30 randomised to 20 mg alteplase or placebo) and the RESTORE-MI trial (Australia New Zealand Clinical Trials Registry; Number 12618000778280) of up to 800 patients (IMR > 32 randomised to low-dose tenecteplase or placebo). In RESTORE-MI, a large phase 3 double-blinded placebo-controlled multicentre randomised trial, initially 243 patients will be recruited in a 1:1:1 fashion (one-third systemic tenecteplase dose, one-sixth systemic tenecteplase dose, and placebo) as part of a dose-finding and cardiac MRI study to evaluate infarct size at 6 months. Subsequently, after determining the optimal tenecteplase dose with a significant infarct size reduction but no substantial predisposition to adverse events such as bleeding, a larger study of 800 randomised patients (with IMR > 32 post-PCI) will aim to determine whether low-dose tenecteplase improves cardiovascular mortality and heart failure rehospitalisation at 24 months compared to placebo.

### 4.10. Novel and/or Future Therapies

A plethora of less well-known prospective and experimental therapies have also emerged recently as potential targets of coronary MVO but remain to be comprehensively evaluated in large clinical trials.

It is well established that NETs are induced during ischaemia-reperfusion and distal microembolisation post-PCI and correlate with infarct size [26,28], and that NET-mediated microthrombosis contributes to myocardial “no-reflow” and MVO. Compared to controls in a rat model, a combination of DNase1 and recombinant tissue-type plasminogen activator (rt-PA) reduced NET density (*p* < 0.001) and “no-flow” area (*p* < 0.05) in the ischaemic region, reduced infarct size (*p* < 0.001) after 3 h of reperfusion, and ameliorated LV remodelling (*p* < 0.05) [119]. Further studies of this therapy in humans are required to determine whether this approach can attenuate MVO.

ITF-1697 is a C-reactive protein-derived tetrapeptide which, based on pre-clinical studies, was proposed to reduce reperfusion injury to hence MVO. However, in a randomised dose-finding study of 402 patients undergoing primary PCI, there was no difference in TIMI flow, cTFC, MBG, ST-elevation resolution, enzymatic infarct size, or clinical outcome between the placebo or treated patients [120]. Enhanced mitochondrial permeability and subsequent mitochondrial swelling drives reperfusion injury and cell death, and attenuation of this has been another proposed target in ACS patients. However, trials of intravenous TR040303 (an inhibitor of mitochondrial permeability transition pores) and Bendavia (a mitochondria-targeted peptide) failed to demonstrate benefit in STEMI patients [1]. Similarly, agents targeting microthrombosis such as pexelizumab [121] (a humanised monoclonal antibody inhibiting C5 complement) and FX06 [122] (a peptide derived from human fibrin) have also failed to improved outcomes, infarct size, or MVO as assessed by CMR.

A multitude of anti-inflammatory therapies have also been evaluated in patients with ACS, but despite early encouraging results in pre-clinical models, to date, none have convincingly demonstrated attenuation of MVO [30]. IL-1 inhibition, with concomitant downstream IL-6 and CRP suppression, has been investigated most extensively in this domain of the literature and is an emerging target in heart failure management. In the CANTOS trial [123] of 10061 prior myocardial infarction patients with a high-sensitivity CRP level ≥ 2 mg/L, canakinumab (a monoclonal antibody targeting IL-1ß) therapy reduced recurrent cardiovascular events (HR 0.85, *p* = 0.021) over median 3.7 years follow-up, and there was a dose-dependent reduction in heart failure hospitalisation and heart failure-related mortality (*p* = 0.037) [124], although MVO specifically was not evaluated. A variety of experimental and smaller clinical trials of colchicine (an anti-mitotic drug which inhibits NLRP3 inflammasome activation) and anakinra (recombinant IL-1 receptor antagonist) in ACS patients have also demonstrated favourable heart failure-related endpoints such as infarct size, the incidence of adverse remodelling, quality of life scores, rehospitalisation rates, and peak aerobic exercise capacity [31,125]. In particular, in the VCUART3 trial of 99 STEMI patients [125,126], anakinra significantly reduced the systemic inflammatory response compared to placebo, and this translated into a reduced incidence of death and new-onset heart failure (*p* = 0.046) or heart failure hospitalisation (*p* = 0.011), although again there was no data with respect to MVO.

The CHILL-MI study of 120 STEMI patients aimed to evaluate the cardioprotective effects of hypothermia using a combination of cold saline rapid infusion and endovascular cooling starting pre-PCI and up to 1 h after reperfusion [127]. Here, there was no difference between the treatment arm and controls in CMR-assessed infarct size as a percent of myocardium at risk (*p* = 0.15), but there was a lower incidence of heart failure (3% vs. 14%, *p* < 0.05) and a possible favourable effect in early anterior STEMI patients (*p* < 0.05) on subgroup analysis, but these need further confirmation in larger trials.

Emerging data suggest that severe MVO and IMH post-PCI, with residual myocardial iron deposition, can lead to ongoing inflammation and adverse LV remodelling. Experimental studies suggest deferoxamine iron chelation may limit ROS generation, and hence reduce ischaemia-reperfusion injury and subsequent infarct size. In a small study of 60 STEMI patients, however, there was a significant reduction in plasma F(2)-isoprostane levels post-PCI in the treated cohort (*p* = 0.04), indicating amelioration of oxidative stress, but no difference in CMR-determined infarct size (*p* = 0.73) versus placebo [128]. Importantly, this study enrolled all unselected STEMI patients and therefore it is possible that future studies selecting only patients with evidence of IMH on CMR may provide a more favourable result.

Finally, temanogrel is an investigational selective 5-HT2A receptor inverse agonist designed to inhibit serotonin-mediated amplification of platelet aggregation and vasoconstriction. The upcoming ARENA study (www.clinicaltrials.gov accessed 14 July 2021; Unique Identifier NCT04848220) in patients undergoing PCI will aim to determine whether temanogrel is a safe and effective treatment of MVO, with the primary outcome being a change in IMR.

## 5. Conclusions

The crucial role of the microcirculation in determining short-term and long-term prognostic clinical outcomes after an ACS is manifest in the literature and continues to emerge as an untapped frontier for potential future therapies. Unfortunately, efforts to date aimed at improving outcomes in patients with MVO have been met with limited success, most likely because it is a multifactorial process with several interdependent underlying pathophysiological mechanisms. There is clearly an unmet need to develop novel strategies to maintain or restore microvascular perfusion in ACS patients, and these strategies require further investigation in large well-powered clinical trials with hard endpoints.

## Figures and Tables

**Figure 1 cells-10-02188-f001:**
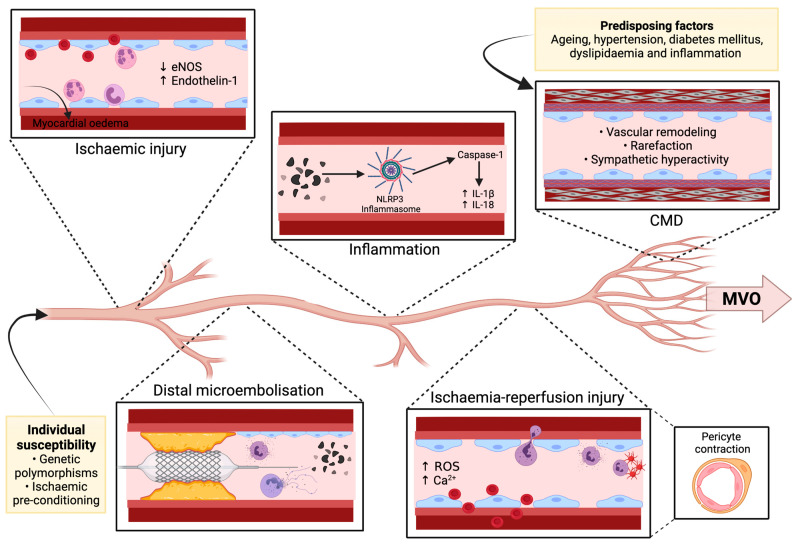
Pathophysiology of MVO. Various genetic polymorphisms and the absence of ischaemic pre-conditioning predispose coronary microcirculation to injury. Plaque disruption, whether spontaneous or PCI-induced, results in distal microembolisation of cellular debris and augmentation of the inflammatory cascade, particularly neutrophil activation. Ischaemic injury to the microcirculation causes endothelial damage and extravasation of erythrocytes and inflammatory cells, resulting in myocardial oedema and luminal narrowing. Moreover, ischaemic injury promotes vasoconstriction by decreasing eNOS release and increasing endothelin-1 release, further limiting microvascular flow. Triggered by inflammatory debris, the NLRP3 inflammasome activates caspase-1 thereby mediating the cleavage of proIL-1b and proIL-18 into their active forms. Reperfusion after a prolonged ischaemic period further potentiates the inflammatory response by augmenting leukocyte recruitment, ROS generation, neutrophil-platelet aggregate formation, and calcium release. Ischaemia-reperfusion injury also induces pericyte contraction and intramyocardial haemorrhage which further limit microvascular flow. Increasing age, hypertension, diabetes mellitus, dyslipidaemia, and inflammation all predispose to CMD, which is characterised by perivascular fibrosis, smooth muscle cell and endothelial dysfunction, vascular remodelling, rarefaction, and sympathetic hyperactivity. These interdependent mechanisms form the underlying pathogenesis of MVO. Abbreviations: Ca^2+^, Calcium; CMD, Coronary microvascular dysfunction; eNOS, Endothelial nitric oxide synthase; MVO, Microvascular obstruction; ROS, Reactive oxygen species.

**Table 1 cells-10-02188-t001:** Classification of CMD [12].

Type	Clinical Setting	Pathogenesis
1—no myocardial diseases or obstructive coronary artery disease	Microvascular angina	Endothelial dysfunction Smooth muscle cell dysfunction Vascular remodeling
2—myocardial diseases	Hypertrophic cardiomyopathy Aortic stenosis Dilated cardiomyopathy Amyloidosis Myocarditis	Smooth muscle cell dysfunction Vascular remodeling Extramural compression
3—obstructive coronary artery disease	Stable angina or acute coronary syndrome	Endothelial dysfunction Smooth muscle cell dysfunction Luminal obstruction
4—iatrogenic	Percutaneous coronary intervention or coronary artery bypass grafting	Luminal obstruction Autonomic dysfunction

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
