# Peer review of "Acute Coronary Syndromes (ACS)—Unravelling Biology to Identify New Therapies—The Microcirculation as a Frontier for New Therapies in ACS"

_cells, 2021, doi:10.3390/cells10092188_

Round 1

Reviewer 1 Report

Vaidya and colleagues wrote a well written review article on microvascular dysfunction. The authors reviewed the pathophysiology well. The format and flow is very similar to the review article written by Niccoli et al in 2009. 

The review on diagnostic investigations for microvascular dysfunction is appropriate. Previous studies have also shown the utility of intracoronary ECG with infarct size and microvascular obstruction. Perhaps, the author may want to include a small paragraph on this for completeness.

The review on all the past, present and future therapies for microvascular dysfunction is excellent. 

Author Response

Thank you for the kind comments. As suggested, we have added a small paragraph (lines 254 to 260) in the revised manuscript on the role of intracoronary ECG in diagnosing MVO.

Reviewer 2 Report

The crucial role of the microcirculation in determining short-term and long-term prognostic clinical outcomes after an acute coronary syndrome (ACS) is manifest in the literature and continues to emerge as an untapped frontier for potential future therapies. The manuscript entitled “Acute Coronary Syndromes (ACS) – Unravelling Biology to Identify New Therapies - The Microcirculation as a Frontier for New Therapies in ACS” provide a detailed summary of the pathogenesis of coronary microvascular obstruction (MVO), diagnosis of MVO during evolving ACS, potential therapeutic strategies and their limitations, and an evaluation of future opportunities on the horizon to mitigate MVO in ACS patients.

I recommend being accepted in present form.

Author Response

Thank you for the generous feedback.

Reviewer 3 Report

Vaidya et al. provide an overview and discussion on the pathogenesis and prognostic significance of coronary microvascular obstruction (MVO) in acute coronary syndromes (ACS) as well as therapeutic approaches targeting MVO. This manuscript addresses an important topic associated with an increased risk of adverse outcome of patients with ACS treated invasively, particularly those with acute STEMI. Both the rationale and scientific content of the manuscript are valuable. In general, the manuscript is well written and provides interesting information. However, I have to indicate that some significant aspects associated with the mechanisms and clinical effects of MVO are missing and, in my opinion, deserve special attention in the revision.

  1. The authors have summarized pretty well the existing results of basic research and clinical studies on the role of MVO for myocardial injury. The authors also have indicated multiple times (e.g., L. 119, 141-143, 266-267, 484), though briefly, that inflammation in the course of ACS may be linked with MVO which can contribute to further myocardial injury. However, inflammation has not been included in the section of the manuscript on the mechanisms underlying MVO and myocardial injury (“Pathogenesis of MVO”). The mechanisms and significance of cardiac inflammation, which can contribute to cardiac fibrosis and development of cardiac dysfunction/remodeling and heart failure (HF), is currently a subject of extensive experimental and clinical investigation in various types of ACS. Evidence from basic research and clinical studies also indicates the associations between MVO, infarct size, inflammation and left ventricular (LV) dysfunction, LV remodeling and long-term HF post-acute STEMI that was treated invasively. I suggest adding a separate subsection in the section “Pathogenesis of MVO” that will focus on this topic and including at least a few references describing the results of basic and clinical studies which provided the proof on the relationships between MVO, inflammation, myocardial injury, cardiac dysfunction and/or remodeling and HF in the long-term follow-up post-ACS. Example relevant references that would fit well are:

Toldo, S., Abbate, A. The NLRP3 inflammasome in acute myocardial infarction. Nat. Rev. Cardiol. 2018, 15, 203–214.

Ørn, S., et al. C-reactive protein, infarct size, microvascular obstruction, and left-ventricular remodelling following acute myocardial infarction. Eur. Heart J. 2009, 30, 1180–1186.

Świątkiewicz, I., et al. Value of C-reactive protein in predicting left ventricular remodelling in patients with a first ST-segment elevation myocardial infarction, Med. Inflamm. 2012, 2012: 250867.

Swiatkiewicz, I., et al. Enhanced inflammation is a marker for risk of post-infarct ventricular dysfunction and heart failure. Int. J. Mol. Sci. 2020, 21, 807; doi:10.3390/ijms21030807.

Świątkiewicz, I., et al. C-Reactive Protein as a Risk Marker for Post-Infarct Heart Failure over a Multi-Year Period. Int. J. Mol. Sci. 2021, 22, 3169, doi.org/10.3390/ijms22063169.

  1. Because the recent clinical studies provided the evidence on the effectiveness of anti-inflammatory treatments in post-infarct patients and patients with acute STEMI for reducing infarct size and clinical outcome, I would suggest adding a subsection on the current status and future perspectives regarding the role of anti-inflammatory therapies for reducing a risk of MVO in patients with ACS in the section “Therapeutic approaches in MVO”. Example relevant references that would fit well are:

Abbate, A., et al. Interleukin-1 and the Inflammasome as Therapeutic Targets in Cardiovascular Disease. Circ. Res. 2020, 126, 1260–1280.

Abbate, A. et al. Interleukin‐1 Blockade Inhibits the Acute Inflammatory Response in Patients With ST‐Segment–Elevation Myocardial Infarction. J. Am. Heart Assoc. 2020, 9; doi.org/10.1161/JAHA.119.014941.

Everett, B.M., et al. Anti-Inflammatory Therapy With Canakinumab for the Prevention of Hospitalization for Heart Failure. Circulation 2019, 139, 1289–1299.

Author Response

Thank you for the valuable comments and suggestions. We have included a section on Inflammation in the Pathophysiology of MVO in ACS (lines 149 to 170) and have amended Figure 1 to reflect this inclusion as well. Furthermore, we have addressed the role of anti-inflammatory therapies in MVO as recommended in the Novel and/or future therapies section (lines 511 to 525).

Round 2

Reviewer 3 Report

The authors adequately addressed my comments.